# Anti-Cytokine Drugs in the Treatment of Canine Atopic Dermatitis

**DOI:** 10.3390/ijms262210990

**Published:** 2025-11-13

**Authors:** Agnieszka Wichtowska, Małgorzata Olejnik

**Affiliations:** Department of Basic and Preclinical Sciences, Faculty of Biological and Veterinary Sciences, Nicolaus Copernicus University, 87-100 Torun, Poland; awichtowska@umk.pl

**Keywords:** canine atopic dermatitis, cytokines, lokivetmab, oclacitinib, ilunocitinib, ciclosporin, immunotherapy

## Abstract

Canine atopic dermatitis (cAD) is a chronic, pruritic, inflammatory skin disease with complex immunopathogenesis involving dysregulated cytokine networks. In recent years, targeted therapies have transformed the management of cAD by directly or indirectly modulating cytokine activity. Lokivetmab, a monoclonal antibody neutralizing interleukin-31, represents a breakthrough in veterinary dermatology, providing rapid and sustained reduction in pruritus with a favorable safety profile. Janus kinase inhibitors, including oclacitinib and the newer ilunocitinib, act downstream by blocking cytokine signal transduction, offering effective control of both acute and chronic phases of disease. Ciclosporin, a calcineurin inhibitor, remains a valuable immunosuppressant for long-term cAD management, while topical tacrolimus provides localized benefits. Together, these therapies mark a paradigm shift from non-specific immunosuppressants to precision medicine. In this context, precision medicine refers to therapeutic strategies that selectively target key cytokines or intracellular signaling pathways central to the pathogenesis of cAD, such as IL-31 or the JAK/STAT axis. Unlike traditional immunosuppressants such as glucocorticoids, which exert broad and non-selective immune suppression, these agents modulate defined molecular mechanisms, thereby improving efficacy and minimizing adverse effects. Consequently, they enable improved quality of life for affected dogs and their owners. Future strategies will likely focus on patient stratification and personalized approaches based on immunological endotypes.

## 1. Introduction

Atopic dermatitis (AD) is one of the most common chronic inflammatory skin diseases in dogs. In a study at the Small Animal Clinic at the University of Montreal, 18.8% of dogs presented with dermatological disorders, of which 12.7% were diagnosed with atopic dermatitis. It is estimated that up to 30% of the dog population may be affected [1]. According to the International Committee on Allergic Diseases of Animals (ICADA) definition (2023), the disease is a chronic, usually pruritic, T-cell-dependent dermatitis resulting from epidermal barrier abnormalities, allergen sensitization, and microbial dysbiosis [2]. Clinically, AD is characterized by persistent pruritus, erythema, lichenification, and frequent secondary bacterial or fungal infections [3]. Constant scratching and licking of the skin often results in self-induced alopecia. Moreover, the skin commonly becomes secondarily infected, further exacerbating the itch [4]. The severity and distribution of lesions vary over time and differ between individuals, making diagnosis and treatment more challenging [5].

Similarly to humans, canine AD, results from a complex interplay of genetic predisposition, environmental influences, and immune response disturbances. It is a heterogeneous, multi-axial immune disease, which complicates the development of effective therapeutic approaches [6,7].

### 1.1. Immunopathogenesis of Canine Atopic Dermatitis

Until recently, canine atopic dermatitis (cAD) was considered a Th2-dominated immune disorder. However, current understanding reveals a more complex picture. Imbalances between cytokines and abnormal interactions between immune cells and keratinocytes are increasingly recognized as central to the pathogenesis [8]. Cytokines regulate inflammatory processes, determine T-lymphocyte response profiles, and modulate epidermal barrier functions. In lesional skin from dogs with AD, in addition to Th2 responses, Th1 (IFN-γ, IL-2, IL-12, IL-15, IL-18), Th17 (IL-17, IL-23) and Th22 (IL-22) axes are activated, along with regulatory cytokines such as IL-10 and TGF-β [8]. Additional cytokines and mediators, including IL-7, IL-34, IL-36, macrophage migration inhibitory factor (MIF), and neurogenic factors (NGF, substance P, periostin) have also been implicated [8].

Marked inter-individual variability is observed: some dogs demonstrate a “pure” Th2 profile, while others show mixed immune responses [9]. The dynamic nature of the disease results in Th2 cytokines dominating acute lesions, while Th1 and Th22 responses predominate in chronic lesions [6,10]. The cytokine system in cAD is dynamic and depends on the disease phase and patient phenotype; taken together, cAD can no longer be defined as a simple “Th2-dependent disorder” (Figure 1) [8]. The schematic diagram was prepared by the authors to summarize key immunological pathways described in the literature, integrating data from studies on Th2, Th1, Th17, and Th22 cytokine axes in canine AD.

The development of cAD involves the activation and differentiation of T and B lymphocytes under the influence of environmental allergens [6]. Epidermal barrier defects permit allergen penetration. Keratinocytes then release “alarmin” cytokines (TSLP, IL-25, IL-33), which activate dendritic cells and ILC2s. In response, dendritic cells present antigen to T lymphocytes and promoting differentiation toward the Th2 phenotype [11]. In healthy skin, lymphocytes are sparse, while in atopic lesions their numbers increase markedly. Production of Th2 cytokines induces IgE synthesis; eosinophil recruitment occurs through IL-5, and mast cell activation is supported by IL-9 [8]. IL-31 is particularly important in the pathogenesis of pruritus in dogs. Mast cell degranulation and eosinophilactivation contribute to chronic skin damage, which in turn promotes activation of the Th1 axis [12,13].

### 1.2. IL-31 and Other Cytokines in cAD

Among cytokines implicated in cAD, IL-31 plays a key pruritogenic role. This cytokine belongs to the gp130/interleukin-6 family and is produced by cells such as Th2 helper T lymphocytes and cutaneous T cells expressing the cutaneous lymphocyte antigen (CLA) [14]. IL-31 is markedly elevated in pruritic skin diseases; in humans, serum IL-31 levels correlate with the severity of atopic dermatitis [10,15]. Despite the recognized role of IL-31 in itch pathogenesis, only a few studies have measured its serum concentrations in dogs. Its precise role in AD remains unclear. Serum IL-31 levels are higher in atopic dogs than in healthy controls, but correlation with clinical severity is inconsistent [10]. No significant association has been observed between IL-31 expression in skin, either serum concentrations or lesion severity; however, strong IL-31 labeling has been detected in sebaceous glands [13].

The heterodimeric receptor for IL-31 is composed of the IL-31 receptor α (IL-31RA) and the oncostatin M receptor β (OSMRβ). Ligand binding activates downstream cascades including JAK-STAT, MAPK, and PI3K pathways. IL-31 receptors are expressed on various cell types, including keratinocytes, macrophages, and eosinophils, where they regulate immune responses. They are also present on sensory neurons, explaining the neuro-immunological feedback loop of the itch–scratch cycle. Experimental administration of canine IL-31 to laboratory beagles induces an immediate and quantifiable pruritic response. The cytokine is also detectable in most dogs with naturally occurring AD [16].

Increased IL-33 expression has been identified in the skin of dogs with AD, particularly in chronic lesions [17]. IL-33 levels in canine atopic skin correlate with lesion severity and chronicity, such as lichenification and erythema [18].

Another keratinocyte-derived cytokine relevant in cAD is thymic stromal lymphopoietin (TSLP), which is activated by allergens (e.g., house dust mites) and bacteria and is believed to exacerbate disease activity [13]. TSLP is induced by the cell wall component of *Staphylococcus* spp. and most cAD patients exhibit colonization and overgrowth by these bacteria [19]. Increased keratinocyte TSLP expression has also been demonstrated after stimulation with house dust mite allergen extracts [20].

Within the Th2 axis in cAD, IL-4 and IL-13 are crucial during the acute phase, as they promote Th2 polarization and IgE synthesis, while IL-5 recruits and activates eosinophils [21]. In horses with insect bite hypersensitivity (IBH), vaccination against IL-5 significantly reduced allergen-specific IgE and modified cytokine expression (IL-4, IL-5, IL-13, IFN-γ), alleviating clinical signs in that species [22].

Cytokines of the Th1 axis detected in cAD include IFN-γ and IL-12, IL-15, IL-18, which stimulate IFN-γ production, although their precise roles remain undefined [8,23,24].

Cytokines of the Th17 and Th22 axes include IL-17, which may initiate inflammation and recruit neutrophils (though results remain inconclusive), and IL-22, which stimulates keratinocyte proliferation, enhances production of antimicrobial peptides, and may contribute to lichenification [25].

Additional cytokines observed in canine AD include regulatory mediators IL-10 and TGF-β, although their precise roles are not fully clarified. Other candidates such as IL-7, IL-34, and IL-36 have been identified as potential biomarkers [13]. In the case of IL-34, markedly elevated serum concentrations have been demonstrated in dogs with AD, correlating with disease severity. However, IL-34 levels were not reduced by systemic or topical steroid therapy or by oclacitinib [26].

## 2. Characteristics of Anti-Cytokine Drugs Used in cAD

### 2.1. Drugs That Directly Neutralize Cytokines

Despite the high prevalence of atopic dermatitis in dogs, only one veterinary drug is currently available that directly targets a cytokine: lokivetmab. Lokivetmab is a monoclonal antibody, marketed under the trade name Cytopoint (Zoetis Inc., Parsippany, NJ, USA). It is the first monoclonal antibody developed specifically for dogs, indicated for the treatment of pruritus associated with allergic dermatitis and the clinical signs of cAD [27,28].

Lokivetmab specifically binds to canine IL-31 in circulation and interstitial fluid, preventing its interaction with its receptor complex (IL-31RA/OSMRβ). As a result, signal transduction is blocked and the pruritogenic cascade and skin-inflammatory components driven by IL-31 are extinguished (the pruritogenic cascade). Pharmacodynamically, lokivetmab has a rapid onset of action; in experimental models, reduced pruritus was observed as early as 8 h after administration [29,30].

As an IgG antibody, lokivetmab has a long elimination phase typical of monoclonal antibodies (mAb), with a half-life in dogs of approximately 2–3 weeks, dependent on FcRn recycling. Pharmacokinetic data indicate a mean terminal half-life of ~16 days, enabling a monthly dosing schedule [29,31].

In dogs with chronic AD, lokivetmab significantly reduced pruritus (measured by Pruritus Visual Analog Scale, PVAS) and lesion severity (Canine Atopic Dermatitis Extent and Severity Index, CADESI-03) compared to placebo. The most robust effect occurred at a dose of 2.0 mg/kg, with clinical benefits persisting for at least one month. A dose–response relationship was observed, confirming the therapeutic relevance of IL-31 neutralization [29].

Retrospective analysis demonstrated that 90% of dogs improved after the first injection, and 77% maintained long-term efficacy [32]. Lokivetmab is comparable to ciclosporin but with fewer adverse effects [27]. In addition, compared with other antipruritic agents available to date, lokivetmab features a long dosing interval, rapid onset, and no age restriction [33]. However, it cannot be used in dogs weighing <3 kg [EMA product information: community register, 2017].

No effect of lokivetmab on hematologic or biochemical parameters has been observed, and the incidence of adverse events was comparable to placebo. No immediate hypersensitivity reactions occurred in the conducted studies. Possible injection-site discomfort occurred in ~5% of dogs [29]. Adverse events occurred in 8% of dogs and were generally mild—most commonly gastrointestinal signs, transient anorexia, or lethargy [32].

When treating with monoclonal antibodies, there is a risk of decreased efficacy due to the formation of anti-drug antibodies (ADAs). The proportion of dogs with detectable treatment-induced immunogenicity (ADA) in studies ranged from 2.1 to 2.6%. Anti-drug antibodies in affected dogs were detected after the second dose [27,29]. According to the summary of product characteristics (EMA), the drug may induce transient or persistent antibody production. In the study by Moyaert et al. (2017), monthly ADA testing detected ADAs within the first 3 months in 3 dogs (2.1%), of which in 2 dogs the response was transient and without clinical impact [27].

Possible causes of sudden loss of efficacy during long-term treatment may also include the presence of pruritus-mediating cytokines other than IL-31 [32].

Clinical guidelines for antipruritic drugs in dogs with AD emphasize a multimodal approach [34]. With lokivetmab, co-therapy is permitted, which is relevant for clinical practice [29]. For many patients, it is optimal to combine lokivetmab with etiologic treatments (allergen-specific immunotherapy, ectoparasite control), topical anti-inflammatory therapy, and elimination of exacerbating factors. No clinically significant interactions have been demonstrated with commonly used products (endectocides, antibiotics, NSAIDs, vaccines) [35].

### 2.2. Drugs Indirectly Affecting Cytokine Signaling—JAK Inhibitors

#### 2.2.1. Oclacitinib

The introduction of oclacitinib into clinical practice in 2013 (2014 in the EU) revolutionized the control of pruritis, offering a rapid onset of action and flexibility to combine with other forms of etiologic therapy (allergen-specific immunotherapy, ectoparasite control, epidermal barrier care). This agent exemplifies an “anti-cytokine” drug acting indirectly—intracellularly—rather than directly on a given cytokine [36,37].

Oclacitinib is a small-molecule Janus kinase (JAK) inhibitor developed for dogs to rapidly and selectively modulate cytokine signaling involved in the pathogenesis of pruritus and dermatitis, particularly in cAD [37]. It shows preferential selectivity for JAK1, primarily inhibiting cytokines that signal via this pathway (including IL-31, IL-4, IL-13). At higher concentrations, it also inhibits JAK2 and JAK3, but therapeutic doses act mainly on JAK1. This explains its rapid and marked antipruritic effect with relatively limited immunosuppression [37,38,39].

The JAK/STAT axis is a central intracellular signal pathway for many cytokines relevant to Th2 responses (including IL-2, IL-4, IL-6, IL-13, IL-31), and its pharmacologic inhibition reduces the activation of pruritogenic neurons and dampens inflammatory components of the skin immune response [37]. Oclacitinib thus inhibits pathways particularly important in allergy, inflammation, and itch, which directly justifies its efficacy in cAD. The half-maximal inhibitory concentrations (IC50) for JAK1-dependent signals are markedly lower than for cytokines not primarily dependent on JAK1 (e.g., erythropoietin, GM-CSF, IL-12, IL-23—IC50 > 1000 nM), defining the functional selectivity profile and reducing the risk of adverse effects from broad immunosuppression [37].

Oclacitinib has high oral bioavailability (~89%), rapid absorption, and a short plasma half-life (~4–6 h), which determine a twice-daily induction regimen, followed by once-daily maintenance dosing. Oclacitinib is indicated for the treatment of clinical manifestations of atopic dermatitis and for the treatment of pruritus associated with allergic dermatitis in dogs. The recommended regimen uses a dose of 0.4–0.6 mg/kg orally twice daily for up to 14 days, then—if treatment continues—once daily at the same dose range [38,40,41]. Clearance and volume of distribution parameters indicate distribution mainly to body fluids and moderate plasma protein binding. Pharmacokinetically, this explains the observed rapid clinical onset and the need for regular dosing to maintain effect [40].

The importance of blocking the IL-31/JAK pathway for reducing canine itch has been confirmed in experimental models in which intradermal administration of recombinant canine IL-31 produced a delayed, quantifiable pruritic response, whereas administration of oclacitinib inhibited this response. These data reinforce the mechanistic–clinical coherence and provide further support for JAK-inhibition as an antipruritic strategy in cAD [15].

Contraindications include dogs under 12 months or weighing <3 kg. The drug should not be used in animals with signs of immunosuppression, such as hypercortisolism, or in dogs with progressive malignant neoplasia. With chronic exposure, infectious and neoplastic events have been reported. When used concomitantly with other potent immunosuppressants (e.g., oral glucocorticoids, ciclosporin), a careful risk assessment is required; both the FDA and the manufacturer emphasize that such combinations may sum their immunosuppressive effects [38,41,42]. As a JAK1/JAK3-modulating immunoresponse drug, oclacitinib may increase susceptibility to infections, including demodicosis, and—theoretically—may exacerbate existing neoplastic processes. These statements reflect the immunomodulatory mechanism of oclacitinib and the potential risk of opportunistic infections or reactivation of skin parasites. Young dogs are more vulnerable to adverse effects related to immune modulation and infections, and the margin of safety in this group has not been confirmed [42].

Interestingly, although oclacitinib does not have an approved feline indication, clinical studies in populations with feline atopic skin syndrome (FASS) or non-flea, non-food-induced hypersensitivity dermatitis (NFNFIHD) suggest that the drug can reduce pruritus and lesion severity in this species as well. In a randomized trial (oclacitinib vs. methylprednisolone), improvements reached 61% reduction in skin lesions (SCORFAD) and 54% reduction in pruritus (VAS) [43]. However, compared with the glucocorticoid (methylprednisolone), the therapeutic effect was more modest. Feline pharmacokinetic differences (e.g., faster absorption and elimination) necessitate different dosing schedules, and the long-term safety database remains limited [44]. Systematic feline studies suggest higher doses are needed (0.6–1.0 mg/kg q12h) [43,45]. From a clinical perspective, oclacitinib is first-line when the primary goal is rapid pruritus control with good tolerability and the clinical picture suggests a pruritogenic phenotype. Unlike direct ligand neutralizers (e.g., anti-IL-31 mAb), oclacitinib acts “downstream,” at the level of signal transduction for many Th2 cytokines, which may be advantageous in dogs with more complex inflammatory patterns. At the same time—as an immunomodulator—it requires careful patient selection and consideration of contraindications. Best outcomes are achieved when symptomatic pharmacotherapy accompanies consistent allergen and infection control, and treatment is monitored using PVAS and CADESI scales and adjusted according to response and adverse effects. A structured and coherent program reduces relapse risk and the need for dose escalation [41].

#### 2.2.2. Ilunocitinib

Within the class of JAK inhibitors, efforts are underway to optimize selectivity profiles and pharmacokinetics to benefit selected phenotypes of affected dogs. For the last decade, oclacitinib has been the main JAK inhibitor in veterinary dermatology, providing rapid itch suppression. In 2024, the FDA approved a new-generation oral JAK inhibitor for the control of pruritus associated with allergic dermatitis and for the control of AD in dogs aged ≥12 months: ilunocitinib (Zenrelia™ Elanco Animal Health, Greenfield, IN, USA) [46,47].

Ilunocitinib exhibits strong functional affinity for JAK1 and significant activity against JAK2 and TYK2, translating into broader cytokine signal suppression. Inhibition of STAT phosphorylation in pathways dependent on these kinases reduces activation of lymphocytes and cutaneous effector cells and suppresses itch transmission by modulating IL-31 signaling on sensory neurons. This mechanism ensures a rapid antipruritic effect while also impacting the inflammatory component of the disease [48]. Because ilunocitinib has a broader activity profile than oclacitinib (including JAK2 and TYK2 to a greater extent), it likely suppresses a wider spectrum of cytokine signals. This may explain its better long-term efficacy (documented in comparative studies), though it also entails a potentially broader range of adverse effects typical of JAK inhibitors (e.g., immunosuppression, infection risk) [46].

After oral administration, ilunocitinib is rapidly absorbed and reaches therapeutic concentrations that ensure 24 h suppression of target JAK pathways, enabling once-daily dosing, which simplifies treatment relative to earlier JAK inhibitors [49]. The most commonly reported adverse effects in clinical studies and practitioner guidance include vomiting, diarrhea, intermittent apathy/lethargy, and infections (e.g., otitis externa). As with all JAK-class immunomodulators, vigilant monitoring for opportunistic infections is necessary and careful assessment is required in patients with a history of neoplasia [50]. As with other JAK inhibitors, vaccination responses may be reduced. Live vaccines are contraindicated during treatment, and careful planning of immunization schedules is necessary [51,52].

In a randomized head-to-head trial, ilunocitinib was compared with oclacitinib in dogs with cAD. Ilunocitinib provided at least equally rapid pruritus control and, in long-term observation (days 28–112), achieved significantly better control of both PVAS and CADESI-04, which may translate into reduced need for rescue interventions and better durability of effect in some patients. A higher proportion of dogs in the ilunocitinib group achieved clinical remission of pruritus. The safety profiles of both drugs were similar over the trial period. These findings suggest that differences in kinase selectivity and pharmacokinetic profile may translate into an advantage for ilunocitinib in maintaining clinical effects with simpler q24h dosing [46,53]. In clinical practice, choosing between oclacitinib and ilunocitinib should consider disease phenotype, dosing-schedule preferences, prior treatment responses, and the patient’s individual risk profile.

#### 2.2.3. Atinvicitinib

The most recent oral JAK inhibitor approved in the European Union in 2025 is atinvicitinib (Numelvi, MSD Animal Health, Rahway, NJ, USA). It is a highly selective, “second-generation” JAK1 inhibitor, which translates into inhibition of signaling by multiple pruritogenic and pro-inflammatory cytokines (including the JAK1-dependent IL-31/IL-4/IL-13 pathways), while exhibiting ≥10-fold weaker activity against JAK2, JAK3, and TYK2 [54,55]. In this context, atinvicitinib provides a more “targeted” JAK1 blockade that could theoretically limit effects on hematopoiesis and JAK2/JAK3-dependent immunity (e.g., in comparison with oclacitinib or the non-selective ilunocitinib) [56], while maintaining a rapid antipruritic effect [36,46]. In registration studies, no impairment of serological responses to core vaccinations was observed even at threefold the recommended dose, supporting the possibility of safely combining therapy with immunoprophylaxis [55].

### 2.3. Drugs Indirectly Affecting Cytokine Signaling—Immunosuppressants Inhibiting Cytokine Synthesis

#### 2.3.1. Ciclosporin

Ciclosporin has long been a cornerstone immunosuppressant in the management of cAD. By binding to cyclophilin in T lymphocytes, it forms a complex that inhibits calcineurin, thereby blocking transcription of IL-2 and other pro-inflammatory cytokines such as IL-4 and IFN-γ. This prevents T-cell activation and proliferation [57,58,59].

In addition to inhibiting T-cell cytokine synthesis, ciclosporin affects epidermal and neuro-immunological mechanisms. It reduces Langerhans-cell function, eosinophil recruitment and activation, mast-cell degranulation, and cytokine production by keratinocytes. Oral administration of ~5 mg/kg daily significantly reduces pruritus and lesion severity, particularly in chronic cases. Its efficacy is comparable to systemic glucocorticoids but with superior long-term tolerability. Moreover, clinical observations suggest no strict correlation between blood drug concentration and response, which is related to CsA accumulation in the skin and underscores the importance of titrating the dose to clinical effect [57,60].

Contraindications include dogs younger than 6 months or weighing <2 kg, and animals with neoplastic disease. Caution is advised in diabetic patients, as ciclosporin may affect glycemia. Live attenuated vaccines should not be administered during treatment or within two weeks before and after therapy; for inactivated vaccines, administration during treatment is discouraged due to possible inadequate vaccine responses [57,61].

Ciclosporin has been evaluated in combination therapies. For example, co-administration with glucocorticoids accelerates onset of action. An attempt was also made to combine oclacitinib with ciclosporin. In a 3-week beagle study, concomitant administration was well tolerated (diarrhea in two dogs receiving both drugs); efficacy was not evaluated [62].

The efficacy and safety of oral oclacitinib versus ciclosporin in dogs with naturally occurring AD have also been compared. Oclacitinib provided a significantly faster antipruritic effect: by days 1–2 owners perceived a reduction in itch clearly greater than in the ciclosporin group, and by day 28 differences in pruritus reduction were statistically significant in favor of oclacitinib. Adverse events occurred in both groups, but gastrointestinal problems were more common with ciclosporin. Oclacitinib offers rapid onset and is better tolerated with respect to gastrointestinal side effects than ciclosporin, making it a valuable option for rapid pruritus control in dogs with AD [63].

#### 2.3.2. Tacrolimus

Tacrolimus, like ciclosporin, is a calcineurin inhibitor but binds to FK506-binding protein (FKBP-12). The tacrolimus–FKBP-12 complex inhibits calcineurin in the same way as the ciclosporin–cyclophilin complex, thereby suppressing transcription of IL-2, IL-4, and other cytokines [59].

In canine AD, tacrolimus (Protopic, LEO Pharma A/S, Ballerup, Denmark, 0.1% ointment) is used off-label as a topical therapy for cutaneous lesions. By inhibiting antigen presentation and cytokine production in keratinocytes and dendritic cells, tacrolimus strengthens the skin barrier and reduces local inflammation. It is particularly valuable when systemic immunosuppressants are contraindicated or undesirable [64,65]. Tacrolimus is used in dogs with cAD, particularly where lesions are localized (e.g., footpads, skin folds, peri-aural regions) and when systemic immunosuppressants are undesirable, contraindicated, or should be minimized [6].

## 3. Therapeutic Guidelines for Anti-Cytokine Drugs in cAD Treatment

In clinical practice, two modern classes of targeted therapeutics form the cornerstone of symptomatic management of pruritus and cutaneous lesions in dogs with atopic dermatitis (CAD): small-molecule Janus kinase (JAK) inhibitors (e.g., oclacitinib, and more recently ilunocitinib and atincitinib) and monoclonal antibodies against interleukin-31 (IL-31) such as lokivetmab. These modalities differ in their molecular targets, pharmacokinetics, routes of administration, and the extent of evidence available from comparative “head-to-head” studies. Additionally, the calcineurin inhibitor cyclosporine remains an established therapeutic option. Given the comparable antipruritic efficacy of oclacitinib and lokivetmab, treatment selection may be guided by patient profile and owner preference. When rapid clinical improvement is prioritized, lokivetmab—administered as an injection with the fastest onset of action (The first clinical effects may be observed as early as 8–12 h after administration) or oral JAK inhibitors, which achieve clinical effect within 1–2 days, are often preferred. Both oclacitinib and lokivetmab have demonstrated improvement of skin barrier parameters and reduction in disease exacerbations; notably, lokivetmab prevented “flares” when administered prophylactically prior to allergen challenge.

For long-term therapy, cyclosporine—characterized by cumulative tissue concentrations—or lokivetmab with its prolonged effect are generally considered optimal choices, the latter being associated with a more favorable safety profile. Clinical evidence indicates that lokivetmab was not inferior to cyclosporine in reducing pruritus as assessed by owner visual analogue scale (VAS) scores; moreover, lokivetmab provided a significantly faster onset of antipruritic effect, with meaningful reduction already within the first 24 h. Regarding safety, lokivetmab showed a superior tolerability profile compared with cyclosporine.

Treatment strategy should also take into account comorbid conditions and the vaccination schedule. In dogs with impaired immunity, neoplastic disease, or during periods of active immunization, drugs with broader immunosuppressive activity should be avoided. In such cases, lokivetmab appears preferable, as a monoclonal antibody with inherently limited systemic immunomodulation beyond the IL-31 axis and minimal interference with hematopoiesis. In contrast, oclacitinib—by suppressing multiple cytokine pathways—requires cautious use in the context of infectious risk or concomitant immunomodulatory therapy. Observational data from practice suggest that oclacitinib treatment may reduce the need for antibiotic therapy compared with other antipruritic strategies, possibly reflecting improved control of the disease and secondary complications.

Equally important for therapy selection are logistical considerations, owner compliance, and ease of administration. Cyclosporine is available both as an oral solution and in capsule form (the latter originally developed for humans), typically administered daily at the initiation of treatment, with potential for reduced frequency once tissue accumulation is achieved. JAK inhibitors are daily oral tablets, whereas lokivetmab is delivered as a subcutaneous injection every 4–8 weeks. The introduction of ilunocitinib and atincitinib expands the portfolio of highly selective oral JAK1 inhibitors; however, the absence of direct comparative trials with lokivetmab leaves open important questions regarding “between-class” differences in efficacy, speed of onset, and long-term safety. In this context, an individualized treatment approach remains the most rational strategy (Table 1) [27,66,67,68,69].

The advent of anti-cytokine therapeutics is displacing the use of glucocorticoids in the management of canine cAD. This is a favorable development, because although glucocorticoids also suppress the synthesis of many cytokines, they act broadly and thereby produce many adverse effects [5]. The ever-expanding arsenal of drugs used to treat cAD enables tailoring of therapy according to the patient’s needs as well as the financial resources and convenience of the owner [60]. Moreover, the emergence of more targeted therapies allows for more rapid therapeutic effects while improving safety profiles [29]. Using drugs that can readily be combined with other therapeutic agents and that do not contribute to an increased risk of secondary infections through amplification of immunosuppression aligns well with a multimodal approach to cAD therapy [70]. Monoclonal antibodies currently appear to represent the most promising therapeutic path, though a potential issue is the development of anti-drug antibodies (ADAs) in some patients [71]. Nevertheless, in the future, selective blockade of individual cytokines without more systemic disruption of homeostasis appears to be the optimal strategy for treating cAD [72]. In preclinical literature, there are reports that certain human mAbs against IL-6 or IL-6R can bind or influence homologous receptors in dogs in vitro, and conceptual works suggest that IL-6 blockade could have applications in veterinary medicine, particularly in inflammatory diseases [73,74]. In veterinary medicine, the field of biologic therapy is rapidly evolving—many projects are currently in preclinical or clinical development [75].

A schematic summary of the immunopathogenesis of cAD and the principal cytokine targets of therapeutic intervention is presented in Figure 2.

## 4. Discussion—Limitations and Perspectives

Despite remarkable progress in targeted immunotherapy for canine atopic dermatitis (cAD), several aspects of disease variability and pharmacologic response still require further investigation. The following considerations summarize current limitations and perspectives for the use of anti-cytokine drugs in cAD.

### 4.1. Breed-Related and Individual Variability

Clinical responses to anti-cytokine therapy may be influenced by breed-specific immune phenotypes, genetic predispositions, and epidermal barrier characteristics. Substantial differences in disease prevalence and clinical expression among breeds have been well documented, suggesting the existence of distinct immunological endotypes [4,6,7]. Although pivotal trials of lokivetmab and oclacitinib included multiple breeds and consistently demonstrated significant mean efficacy [27,35], they were not designed or powered to assess inter-breed differences. The potential impact of genetic background on cytokine receptor expression, JAK signaling activity, or drug disposition remains an open field of research. Future prospective studies integrating genomic or transcriptomic profiling may help to identify breed-related biomarkers predictive of therapeutic response.

### 4.2. Pharmacokinetic Variability and Influencing Factors

The pharmacokinetic (PK) profile of monoclonal antibodies such as lokivetmab is largely determined by FcRn-mediated recycling, resulting in a mean terminal half-life of approximately 16 days and allowing a convenient monthly dosing schedule [29,31]. Interindividual variability in drug exposure may arise from differences in body weight, metabolic rate, and immunogenicity. The occurrence of anti-drug antibodies (ADAs), reported in about 2–3% of treated dogs, may transiently reduce circulating drug levels [27]. Physiological or pathological factors altering plasma protein turnover—such as protein-losing enteropathy or nephropathy—may also influence mAb clearance [71]. For small-molecule JAK inhibitors, variability may reflect differences in gastrointestinal absorption, hepatic metabolism, or pharmacogenomic polymorphisms in cytochrome P450 enzymes and drug transporters. These aspects warrant further study to refine dosing strategies and optimize therapeutic predictability in clinical settings.

### 4.3. Cytokine Heterogeneity, Allergen Dependence, and Personalized Therapy

The cytokine network in cAD is heterogeneous and dynamically modulated by allergen type, disease phase, and individual immune regulation. Acute lesions are characterized by Th2-dominant cytokines such as IL-4 and IL-13, while chronic lesions exhibit Th1 (IFN-γ, IL-12) and Th22 (IL-22) signatures [8,13,21]. Allergen-specific immunotherapy has been shown to modulate these responses, shifting the immune balance toward regulatory or Th1 patterns [11]. Consequently, therapeutic strategies should be tailored according to immunological phenotype: anti-IL-31 antibodies such as lokivetmab may be optimal for pruritus-dominant phenotypes, while JAK inhibitors provide broader cytokine modulation suitable for mixed or chronic inflammatory patterns. Combining biologics with etiologic therapies (ASIT, infection control, and barrier repair) aligns with precision-medicine principles and may improve long-term outcomes. Further integration of immunoprofiling into clinical trials will enhance the development of personalized, mechanism-based treatment algorithms in veterinary dermatology.

## 5. Conclusions

The advent of anti-cytokine drugs has fundamentally changed the therapeutic landscape of canine atopic dermatitis. These agents provide rapid and reliable relief from pruritus while reducing the long-term reliance on broadly acting immunosuppressants. Their major strengths lie in the ability to achieve targeted, cytokine-specific modulation with improved safety and owner compliance.

Lokivetmab, the first veterinary monoclonal antibody, represents a milestone by offering a safe and durable antipruritic effect with minimal systemic immunomodulation. The successive introduction of oral JAK inhibitors—oclacitinib, ilunocitinib, and atinvicitinib—has further diversified the therapeutic armamentarium, allowing flexible, patient-tailored approaches that match disease phenotype and clinical context.

Nevertheless, as highlighted in the preceding section, future research should aim to define immunological endotypes, elucidate pharmacokinetic variability, and identify biomarkers predictive of treatment response. Integrating such precision-medicine principles into veterinary dermatology will enable the development of more individualized, mechanism-based treatment strategies, ultimately improving long-term disease control and quality of life for affected dogs and their owners.

## Figures and Tables

**Figure 1 ijms-26-10990-f001:**
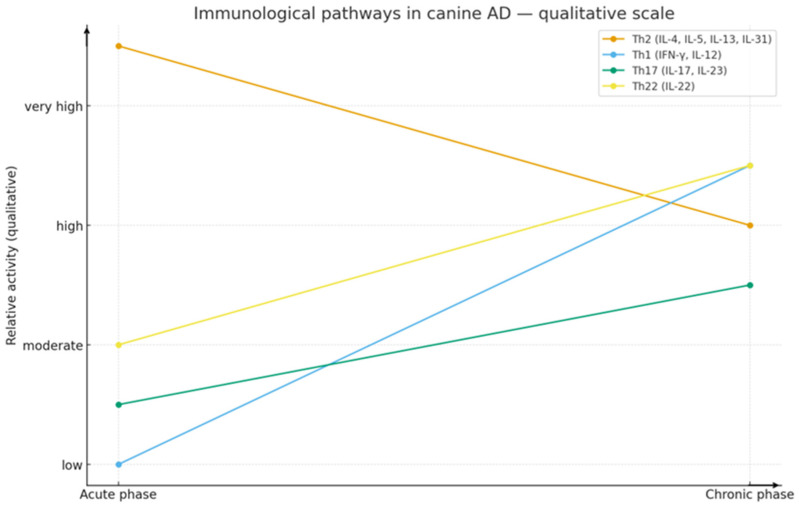
Conceptual illustration of the immune mechanisms underlying canine atopic dermatitis. The diagram was designed by the authors to summarize current knowledge on cytokine-mediated interactions between keratinocytes, dendritic cells, and T lymphocytes in cAD, based on data reported in the recent literature [8,10,11,12,13]. The diagram was generated using Canva https://www.canva.com (accessed on 25 September 2025).

**Figure 2 ijms-26-10990-f002:**
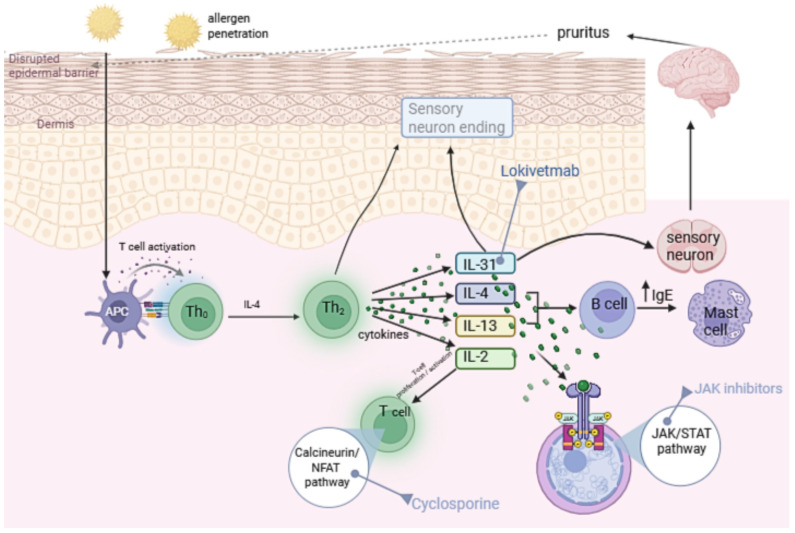
The figure was prepared by the authors. It conceptually summarizes the immunopathogenesis of canine atopic dermatitis and pharmacologic targets of anti-cytokine therapies, based on published data on Th2 cytokines, IL-31 signaling, and JAK/STAT and calcineurin/NFAT pathways.

**Table 1 ijms-26-10990-t001:** Comparison of Anti-Cytokine and Immunomodulatory Drugs.

Drug	Target/Mechanism	Formulation/Route	Typical Dose & Schedule	Main Advantages	Limitations/Adverse Effects
Lokivetmab	Neutralizes IL-31 directly (mAb)	Injectable (SC)	2 mg/kg q4w (monthly)	Rapid onset, long dosing interval, minimal side effects	Weight limit (>3 kg), GI signs, rare ADAs
Oclacitinib	Selective JAK1 inhibitor (IL-31, IL-4, IL-13)	Oral tablets	0.4–0.6 mg/kg BID × 14 d → QD	Very fast pruritus relief, flexible dosing, convenient	Contraindicated <12 mo, infection/neoplasia risk, daily dosing
Ilnocitinib	JAK1/JAK2/TYK2 inhibitor (broader)	Oral tablets	Once daily	Strong itch and lesion control, simple q24h dosing	Broader immunosuppression, infection risk
Atinivicytinib	Next-gen JAK inhibitor (JAK1/3, TYK2)	Oral tablets	Once daily (investigational)	Potent, broad cytokine modulation, promising efficacy	Limited clinical data; possible immunosuppression, infection risk
Ciclosporin	Calcineurin inhibitor → ↓ IL-2, IL-4, IFN-γ synthesis	Oral solution; Oral capsules	≈5 mg/kg QD	Effective long-term control, steroid-sparing	Slow onset, GI side effects, contraindications
Tacrolimus	Calcineurin inhibitor (topical)	Topical ointment	Local application (0.1% oint.)	Useful for localized lesions, avoids systemic exposure	Limited to focal lesions, local irritation

## Data Availability

No new data were created or analyzed in this study. Data sharing is not applicable to this article.

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
