# Peer review of "Anti-Cytokine Drugs in the Treatment of Canine Atopic Dermatitis"

_ijms, 2025, doi:10.3390/ijms262210990_

Round 1

Reviewer 1 Report

Comments and Suggestions for Authors

The title may be changed to “Anti‑Cytokine Drugs in Canine Atopic Dermatitis Treatment” or a similar title.

Abstract line 19: “Together, these therapies mark a paradigm shift from non‑specific immunosuppressants to precision medicine”. The point of precision medicine in that respect is unclear and was not sufficiently or clearly described in the manuscript.

The method by which figure 1 was constructed should be clearly explained.

Line 122: for the title 2.1. Drugs that Directly Neutralize Cytokines, there is no need for subtitles if only one type of drugs is discussed.

A graphical summary or a figure summarizing mechanisms of Atopic dermatitis and mechanisms of action of the discussed drugs is recommended.

Line 419: It is not clear what is meant by “precision medicine” here as this point or idea was not discussed.

Author Response

We sincerely thank Reviewer 1 for the careful evaluation of our manuscript and for the constructive comments and suggestions. We greatly appreciate the reviewer’s insightful feedback, which has helped us to improve the clarity, precision, and scientific quality of the paper. All suggestions were carefully considered and incorporated into the revised version of the manuscript. Below, we provide a detailed point-by-point response to each comment.

Comments 1: “The title may be changed to ‘Anti-Cytokine Drugs in Canine Atopic Dermatitis Treatment’ or a similar title.”

Response 1: We appreciate this valuable suggestion. The title has been revised to improve clarity and to better reflect the therapeutic focus of the manuscript. The new title reads: “Anti-Cytokine Drugs in the Treatment of Canine Atopic Dermatitis.” This formulation follows standard scientific phrasing and aligns with the manuscript’s emphasis on pharmacologic mechanisms and clinical applications of anti-cytokine therapies in canine atopic dermatitis.

Comments 2: "Abstract line 19: “Together, these therapies mark a paradigm shift from non-specific immunosuppressants to precision medicine”. The point of precision medicine in that respect is unclear and was not sufficiently or clearly described in the manuscript."

Response 2: We appreciate this comment. The concept of precision medicine has been clarified and expanded throughout the manuscript. Specifically, the revised Abstract now defines precision medicine in the context of canine atopic dermatitis (cAD) as “therapeutic strategies that selectively target key cytokines or intracellular signalling pathways central to the pathogenesis of cAD, such as IL-31 or the JAK/STAT axis.” Furthermore, a new section (4. Discussion – Limitations and Perspectives) was added to elaborate on this approach and its future potential in individualized therapy.

Comments 3: "The method by which Figure 1 was constructed should be clearly explained."

Response 3: A detailed explanation of Figure 1’s preparation has been included in the text and figure legend. The figure was created by the authors and integrates data synthesized from multiple recent publications on Th2, Th1, Th17, and Th22 cytokine axes in cAD (Pucheu-Haston et al., 2015; Olivry et al., 2016; Tamamoto-Mochizuki et al., 2024). This clarification appears in the legend immediately following Figure 1 in the revised version.

Comments 4: "Line 122: for the title “2.1. Drugs that Directly Neutralize Cytokines,” there is no need for subtitles if only one type of drugs is discussed."

Response 4: Thank you for noting this. The section has been reformatted accordingly. The redundant subtitle has been removed, and the text now flows directly under “2.1. Drugs that Directly Neutralize Cytokines,” which discusses lokivetmab as the only direct anti-cytokine agent currently approved in veterinary dermatology.

Comments 5: "A graphical summary or a figure summarising mechanisms of Atopic dermatitis and mechanisms of action of the discussed drugs is recommended."

Response 5: Following this suggestion, we added a new graphical summary (Figure 2) illustrating the immunopathogenesis of canine atopic dermatitis and the mechanisms of action of key anti-cytokine and immunomodulatory drugs (lokivetmab, JAK inhibitors and ciclosporin). This visual element provides a concise overview complementing the narrative text and facilitating reader understanding.

Comments 6: "Line 419: It is not clear what is meant by “precision medicine” here as this point or idea was not discussed."

Response 6: This concern has been fully addressed. The concept of precision medicine is now explicitly developed in the newly added Section 4 (Discussion – Limitations and Perspectives). This section discusses heterogeneity in cytokine profiles, breed-related differences, pharmacokinetic variability, and the rationale for personalized, mechanism-based therapeutic selection in cAD. The revised Conclusions also include a bridging paragraph emphasising future directions toward precision and individualised therapy.

Reviewer 2 Report

Comments and Suggestions for Authors

This manuscript clearly presents the immunopathological mechanisms underlying various cytokine changes in CAD, along with the mechanisms of action and clinical applications of cytokine inhibitors and modulators.

It is expected to provide researchers with valuable information for understanding the immunopathology of CAD and the development and understanding of immunomodulatory drugs.

- Clinical applications of CAD involve numerous variables, so additional consideration should be given to differences in clinical outcomes across breeds.

- Additional consideration may also be given to factors that may contribute to differences in PK profiles.

- Cytokine profiles in CAD vary depending on the allergen, and treatment and improvement strategies may vary depending on the course of CAD. Further insight from the author is warranted.

Author Response

We would like to express our sincere gratitude to Reviewer 2 for the thorough review and thoughtful comments on our manuscript. We highly appreciate the reviewer’s positive assessment and the valuable suggestions that contributed to improving the scientific content and clarity of the paper. All comments have been carefully addressed, and appropriate revisions were incorporated into the revised version of the manuscript.

Comments 1: "Clinical applications of CAD involve numerous variables, so additional consideration should be given to differences in clinical outcomes across breeds."

Response 1: A detailed discussion on breed-related and individual variability has been added to the new Section 4.1 (Breed-Related and Individual Variability). This section summarizes current evidence for breed-specific immune and barrier phenotypes, genetic predispositions, and their potential influence on therapeutic response, citing recent literature (Outerbridge & Jordan 2021; Hensel et al. 2024; Drechsler et al. 2024; Moyaert et al. 2017; Van Brussel et al. 2021).

Comments 2: "Additional consideration may also be given to factors that may contribute to differences in PK profiles."

Response 2: Section 4.2 (Pharmacokinetic Variability and Influencing Factors) now addresses interindividual differences in pharmacokinetics of monoclonal antibodies and JAK inhibitors, including the roles of FcRn recycling, body size, metabolic rate, and anti-drug antibodies. Relevant references (Michels et al. 2016; Krautmann et al. 2023; Tabrizi et al. 2006) have been incorporated.

Comments 3: "Cytokine profiles in CAD vary depending on the allergen, and treatment and improvement strategies may vary depending on the course of CAD. Further insight from the author is warranted."

Response 3: A comprehensive discussion has been added in Section 4.3 (Cytokine Heterogeneity, Allergen Dependence, and Personalized Therapy), describing how cytokine expression patterns differ according to allergen exposure and disease phase (Th2 vs. Th1/Th22 dominance). The section also explains how this variability may inform therapeutic choices—e.g., anti-IL-31 antibodies for pruritus-dominant phenotypes versus JAK inhibitors for broader inflammatory control—and highlights the potential of integrating cytokine profiling into clinical decision-making.